# The rise in domestic shigellosis and the genomic characteristics of *Shigella* clones linked to men who have sex with men in Taiwan, 2015–2022

Ying-Shu Liao,[1] Bo-Han Chen,[1] Yu-Ping Hong,[1] You-Wun Wang,[1] Ru-Hsiou Teng,[1] Shiu-Yun Liang,[1] Jui-Hsien Chang,[1] Chi-Sen Tsao,[1] Hsiao Lun Wei,[1] Chien-Shun Chiou[1]

**ABSTRACT**   Since 2015, Taiwan has experienced a notable rise in domestic shigellosis cases, particularly among young adult males. In this study, we aimed to investigate this epidemiological trend through demographic analysis and genomic characterization of bacterial isolates. We analyzed demographic data on shigellosis cases from 2003–2014 to 2015–2022. For cases from 2015–2022, we conducted genomic analyses of *Shigella* isolates using pulsed-field gel electrophoresis (PFGE) and whole-genome sequencing (WGS). Antimicrobial resistance and plasmid profiles were examined to identify genetic determinants of resistance. From 2015 to 2022, there was a noticeable demographic shift in domestic shigellosis cases from females to males and children to young adults. This trend was driven by three multidrug-resistant *Shigella* clones associated with men who have sex with men (MSM): ciprofloxacin-resistant *S. sonnei* (CipR_SSIII), azithromycin-resistant *S. flexneri* 3a (AziR_SF3), and ciprofloxacin-resistant *S. flexneri* 2a (CipR_SF2). The CipR_SF2 clone has become the most prevalent since 2018, responsible for 84.9% of cases in 2021. Genomic analysis revealed that CipR_SF2 isolates are genetically distinct from those involved in MSM-related outbreaks in other countries. These MSM-associated clones showed significantly higher resistance to azithromycin, extended-spectrum cephalosporins (ESCs), and ciprofloxacin. Additionally, 14 extensively drug-resistant isolates were identified, carrying resistance genes for azithromycin and ESCs on IncFII or IncB/O/K/Z plasmids. Our findings indicate that the increase in domestic shigellosis cases in Taiwan from 2015 to 2022 is primarily attributed to the spread of highly resistant MSM-associated *Shigella* clones.

**IMPORTANCE**   The rise of multidrug-resistant (MDR) and extensively drug-resistant (XDR) Shigella strains poses a growing global health threat, particularly among high-risk groups such as men who have sex with men (MSM). This study highlights the increasing prevalence of domestic shigellosis in Taiwan from 2015 to 2022, driven by the emergence of three MDR Shigella clones. These MSM-associated clones exhibit significantly higher resistance to azithromycin, extended-spectrum cephalosporins (ESCs), and ciprofloxacin compared to non-MSM-associated clones. Additionally, 14 extensively drug-resistant isolates were identified, carrying resistance genes for azithromycin and ESCs on IncFII or IncB/O/K/Z plasmids. Genomic analysis reveals that ciprofloxacin-resistant Shigella flexneri 2a (CipR_SF2) has become the most dominant clone, responsible for the majority of shigellosis cases since 2018, and is genetically distinct from strains observed in MSM-related outbreaks in other countries. By elucidating these clones' genetic characteristics and epidemiological trends, this research offers essential data for public health surveillance, helping to inform strategies for controlling the spread of MDR and XDR Shigella infections.

Address correspondence to Chien-Shun Chiou, nipmcsc@cdc.gov.tw; nipmcsc@gmail.com.

The authors declare no conflict of interest.

See the funding table on p. 11.

**KEYWORDS** shigellosis, *Shigella* spp., antimicrobial resistance (AMR), extensively drug resistance (XDR), multidrug resistance (MDR), men who have sex with men (MSM), Taiwan

Shigellosis, an acute gastrointestinal disease caused by the genus *Shigella*, is primarily characterized by diarrhea, fever, and stomach cramps. Globally, shigellosis remains a significant public health concern, contributing to an estimated 164,000 deaths annually, particularly among children under 5 years old in low- and middle-income countries (LMICs) (1). Traditionally, the epidemiology of shigellosis has been dominated by cases in LMICs; however, recent years have seen a notable shift, with increasing outbreaks reported in high-income countries, particularly among men who have sex with men (MSM) (2).

The management of shigellosis has become increasingly challenging due to the rise of antimicrobial resistance (AMR) in *Shigella* species. Traditional first-line drugs such as ampicillin, chloramphenicol, trimethoprim-sulfamethoxazole, and nalidixic acid have lost efficacy due to widespread resistance. Consequently, fluoroquinolones (e.g., ciprofloxacin), extended-spectrum cephalosporins (ESCs, e.g., ceftriaxone), and azithromycin have become the preferred treatments (1). However, the emergence of multidrug-resistant (MDR) and extensively drug-resistant (XDR) *Shigella* strains further complicates treatment, as resistant strains often spread across countries and continents, facilitated by travelers and MSM (3, 4).

In Taiwan, shigellosis has historically been relatively uncommon, with an average incidence of 1.16 cases per 100,000 people from 2003 to 2022. However, since 2015, a significant epidemiological shift has been observed, with the majority of cases being domestically acquired. This change coincides with the emergence of highly resistant strains, including ciprofloxacin-resistant *Shigella sonnei* and azithromycin-resistant *Shigella flexneri* 3a, which are associated with MSM (5, 6). Furthermore, an inter-hospital investigation has revealed a shift toward the predominance of ciprofloxacin-resistant *S. flexneri* 2a strains since 2018, along with the emergence of a few cefotaxime-resistant *S. flexneri* strains (7).

This study aims to analyze the demographic trends of shigellosis in Taiwan between 2003–2014 and 2015–2022, focusing on the genomic characteristics of *Shigella* isolates from 2015 to 2022. Utilizing pulsed-field gel electrophoresis (PFGE) and whole-genome sequencing (WGS), we seek to understand the clonality and genetic determinants of antimicrobial resistance and track the evolution and dissemination of resistant strains. Our findings will enhance understanding of the epidemiological trends and genomic characteristics of MDR and XDR *Shigella* strains in Taiwan, providing insights into their spread and informing public health strategies.

## RESULTS

### Epidemiology of shigellosis

The NIDSS data showed a sharp upward trend in the proportion of domestic shigellosis cases since 2015, with all cases in 2021 being domestically acquired due to the COVID-19 pandemic lockdown (Fig. S1). The proportion of domestic cases increased from 48.6% (954/1,961) during 2003–2014 to 67.3% (846/1,257) during 2015–2022 (Table S1). Additionally, the sex distribution changed, with males accounting for 44.0% (798/1,815) of cases in 2003–2014, increasing to 68.7% (851/1,238) in 2015–2022. Among domestic cases, the proportion of males increased from 53.6% (441/822) to 92.3% (579/627) during the same periods (Fig. S2) while among the imported cases, the proportion of males remained relatively consistent at 34.6% to 29.0%.

The age and sex distribution of domestic shigellosis cases changed significantly between the two periods. From 2003 to 2014, the majority of domestic cases occurred in children (ages 0–9), with males accounting for a smaller proportion of cases across all age groups (Fig. 1A). In contrast, from 2015 to 2022, domestic cases were predominantly

**(A) 2003–2014**

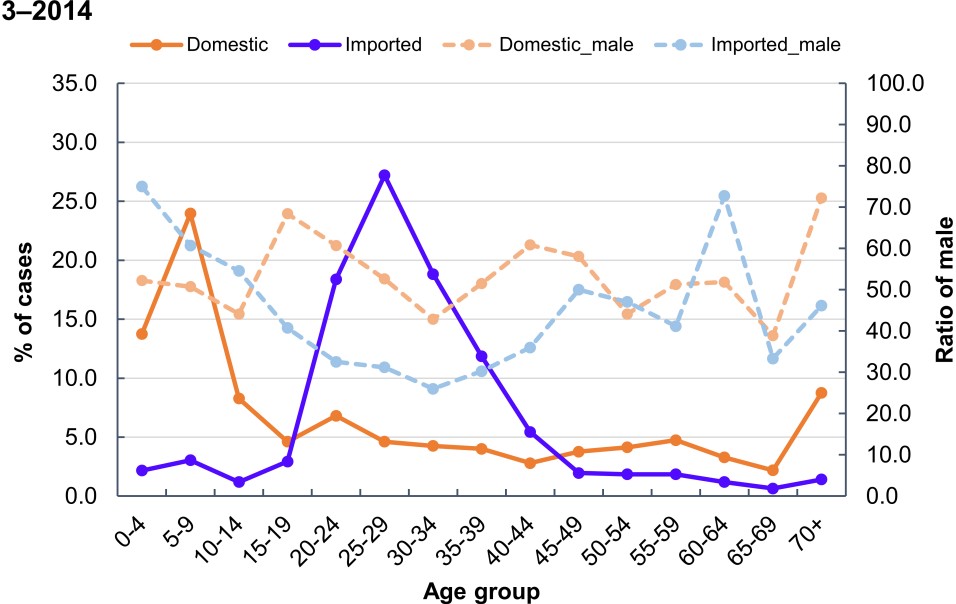

**(B) 2015–2022**

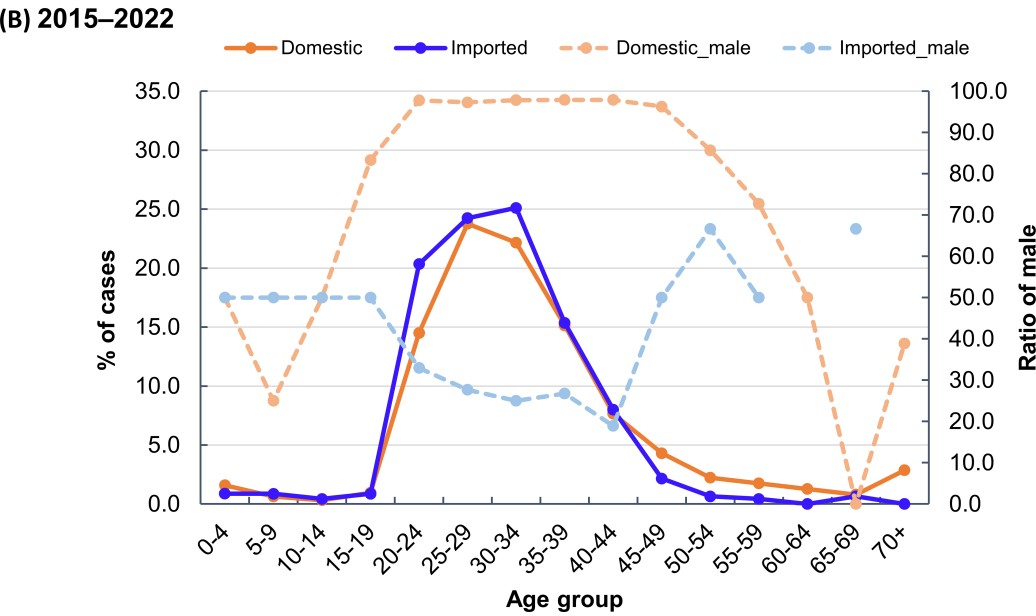

**FIG 1** Distribution of shigellosis cases by age and sex in 2003–2014 (A) and 2015–2022 (B).

observed in young adults (ages 20–39), with males comprising a substantially higher proportion of cases across most age groups, particularly among young adults (Fig. 1B).

### *Shigella* isolates and genogrouping

*S. flexneri* and *S. sonnei* were the most prevalent species identified. From 2015 to 2022, *S. flexneri* infections increased significantly, rising from 38.6% in 2015 to 86.0% in 2021 (Table S2). Clustering analysis of PFGE patterns classified *S. flexneri* isolates into distinct clusters, designated as SF1, SF2, SF3_A, SF3_B, SF4, and SF6, corresponding to *S. flexneri* serotypes 1, 2, 3, 3, 4, and 6, respectively (Fig. S3). SF2 predominantly comprised *S. flexneri* 2a, while the majority of SF3_A and SF3_B were *S. flexneri* 3a. Ciprofloxacin-resistant SF2 isolates were re-assigned to the CipR_SF2 genogroup. SF3_A isolates recovered in

2015 and 2016 were previously identified as belonging to the azithromycin-resistant MSM-outbreak-associated lineage (3, 5), thus SF3_A was renamed AziR_SF3.

*S. sonnei* isolates were grouped into SSII and SSIII, with SSIII further subdivided into three subclusters: SSIII_A, SSIII_B, and SSIII_C (Fig. S4). Subsequent genomic analysis revealed that SSII isolates with WGS data belonged to lineage II, while SSIII isolates were part of lineage III, as designated by Holt et al. (8). Further analysis indicated that SSII isolates corresponded to Mykrobe genotypes 2.7.3 and 2.7.4. SSIII_A isolates were classified into genotypes 3.7.3 and 3.7.6, SSIII_B under genotypes 3.6.1, 3.6.1.1, and 3.6.1.1.1, and SSIII_C under genotype 3.7.29.1.4.1. Most SSIII_B isolates exhibited ciprofloxacin resistance and these ciprofloxacin-resistant isolates were therefore assigned to the CipR_SSIII genogroup to distinguish them from the remaining isolates within the SSIII_B cluster. Accordingly, the SSIII genogroup comprised isolates from SSIII_A, SSIII_B (excluding CipR_SSIII), and SSIII_C.

## Genomic characteristics of *S. flexneri* isolates

Genomic analysis indicated that AziR_SF3 (SF3_A) and SF3_B isolates harbored multiple antimicrobial resistance genes (ARGs) (Table S3). AziR_SF3 isolates typically harbored seven resistance genes: *aadA1*, *blaOXA-1*, *blaTEM-1*, *catA1*, *erm(B)*, *mph(A)*, and *tet(B)*, although two isolates exhibited deletions of some ARGs. AziR_SF3 isolates were closely related, belonging to the same single nucleotide polymorphism (SNP) cluster (PDS000061830.752) and HC20 cluster (HC20_1549), within which strains differ by up to 20 cgMLST loci (Table S3). Four AziR_SF3 strains from 2015 and 2016 were previously confirmed to be linked to an MSM-associated shigellosis outbreak in Taiwan (5). In this study, the phylogenetic analysis using core genome single nucleotide polymorphism (cgSNP) profiles showed that AziR_SF3 isolates clustered with *S. flexneri* 3a isolates from MSM-associated shigellosis outbreaks in Australia (9), Spain (10, 11), and the United Kingdom (12) (Fig. S5). Additionally, two SF3_B isolates (R15.3430 and R18.0082) were found within a cluster of *S. flexneri* 3a isolates from MSM-linked outbreaks in Spain and the United Kingdom (Fig. S5). Isolates within this cluster differed by no more than 116 SNPs.

All SF2 and CipR_SF2 isolates harbored the resistance genes *aadA1*, *sat2*, *dfrA1*, and *tet(B)* (Table S4). CipR_SF2 isolates also possessed two mutations in *gyrA* (S83L and D87N) and one in *parC* (S80I). Some CipR_SF2 isolates carried resistance genes associated with AmpC β-lactamases, such as *blaCMY-2* and *blaDHA-1*, as well as extended-spectrum β-lactamase genes, including *blaCTX-M-14 and blaCTX-M-55*. Additionally, some isolates harbored macrolide resistance genes, including *erm(B)* and *mph(A)*. CipR_SF2 isolates were closely related, belonging two SNP clusters (PDS000121152.7 and PDS000174037.7) and three HC20 clusters (HC20_132319, HC20_158649, and HC20_158933) within the same HC50 cluster (HC50_196) (Table S4). Phylogenetic analysis using cgSNP profiles grouped CipR_SF2 isolates into two clusters, with isolates within each cluster differing by up to 27 and 32 SNPs, respectively (Fig. 2). These two clusters also included isolates from Australia, France, and the United Kingdom, with unknown sources. CipR_SF2 isolates were genetically distant (over 100 SNPs) from isolates associated with MSM-related outbreaks in Australia (9), the Netherlands (13), Spain (10, 11), and the United Kingdom (12) (Fig. 2). In comparison to CipR_SF2 isolates, SF2 isolates were more genetically diverse.

## Genetic characteristics among *S. sonnei* isolates

Genomic analysis revealed that all *S. sonnei* isolates with genotypes 3.6.1, 3.6.1.1, and 3.6.1.1.1 harbored *dfrA1* and *sat2*, while genotypes 3.7.29 (including 3.7.29.1 and 3.7.29.1.4.1), 3.7.3, and 3.7.6 harbored *aadA1*, *dfrA1*, and *sat2* within their chromosomes (Table S5). Most SSIII isolates also carried *aph(3")-Ib*, *aph(6)-Id*, *sul2*, and *tet(A)* on plasmids. Genotypes 3.6.1, 3.7.29.1, 3.7.29.1.4.1, and 3.7.6 exhibited a mutation in *gyrA* (S83L), while genotypes 3.6.1.1 and 3.6.1.1.1 had three mutations: two in *gyrA* (S83L and D87G) and one in *parC* (S80I). Most isolates were found to carry multiple plasmid replicons,

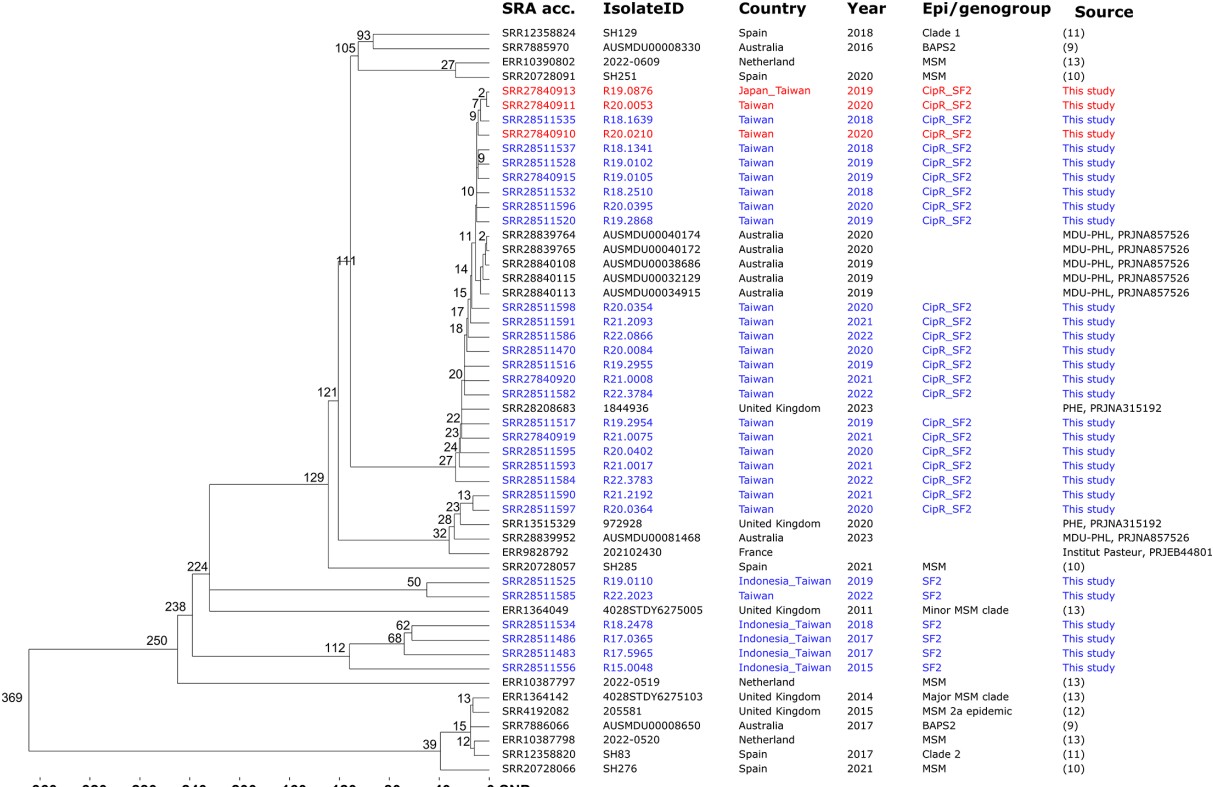

**FIG 2** Phylogenetic tree of *S. flexneri* 2a isolates from Taiwan and other countries. The tree was built using cgSNP profiles with the single-linkage algorithm, using *S. flexneri* 2a strain 2457T as the reference for SNP calling. Taiwanese isolates are highlighted in blue and XDR isolates from Taiwan are highlighted in red.

suggesting the presence of multiple plasmids. The CipR_SSIII group comprised isolates with genotypes 3.6.1.1, 3.6.1.1.1, and one 3.6.1 isolate with ciprofloxacin resistance, which was attributed to the *gyrA* S83L mutation and the carriage of *qnrB4*. Two ciprofloxacin-resistant isolates (C09.0001 and R13.0937), imported from India and Cambodia in 2009 and 2013, respectively, belonged to genotypes 3.6.1.1 and 3.6.1.1.1. All isolates with genotypes 3.6.1.1 and 3.6.1.1.1 belonged to the same HC20 cluster (HC20_385) and SNP cluster (PDS000188704.8).

Phylogenetic analysis revealed that SSIII, including CipR_SSIII isolates, were closely related to strains from Australia (9, 14), Belgium (15), China (16), India/Vietnam (17), the Netherlands (13), Spain (10, 11), and the United Kingdom (4, 12, 18) (Fig. 3). Some of these strains were linked to MSM-associated outbreaks in these countries. Ciprofloxacin-resistant *S. sonnei* associated with MSM in Taiwan was first identified in 2015 (6), and genomic analysis indicated that the outbreak strains recovered in 2015 belonged to genotype 3.6.1.1.

### Epidemiological trend and antimicrobial resistance

The genogroups CipR_SSIII, AziR_SF3, and CipR_SF2 were prevalent between 2015 and 2022, collectively accounting for 42.1% of infections in 2015 and 86.0% in 2021 (Table S2). CipR_SSIII was most prevalent in 2015 and 2016 but became rare after 2018 (Fig. 4). AziR_SF3 predominated in 2017 but was not detected in 2021 and 2022. CipR_SF2 emerged as the most prevalent genogroup from 2018 onward, responsible for 65.0% to 84.9% of total infections between 2019 and 2022. Most cases caused by these three genogroups were domestically acquired (91.9%) and occurred in males (94.5%) (Table 1). Previous studies have indicated that the CipR_SSIII and AziR_SF3 genogroups are associated with the MSM population (5, 6), and our epidemiological data suggest that CipR_SF2 is also likely circulating within the MSM group. In contrast, shigellosis cases

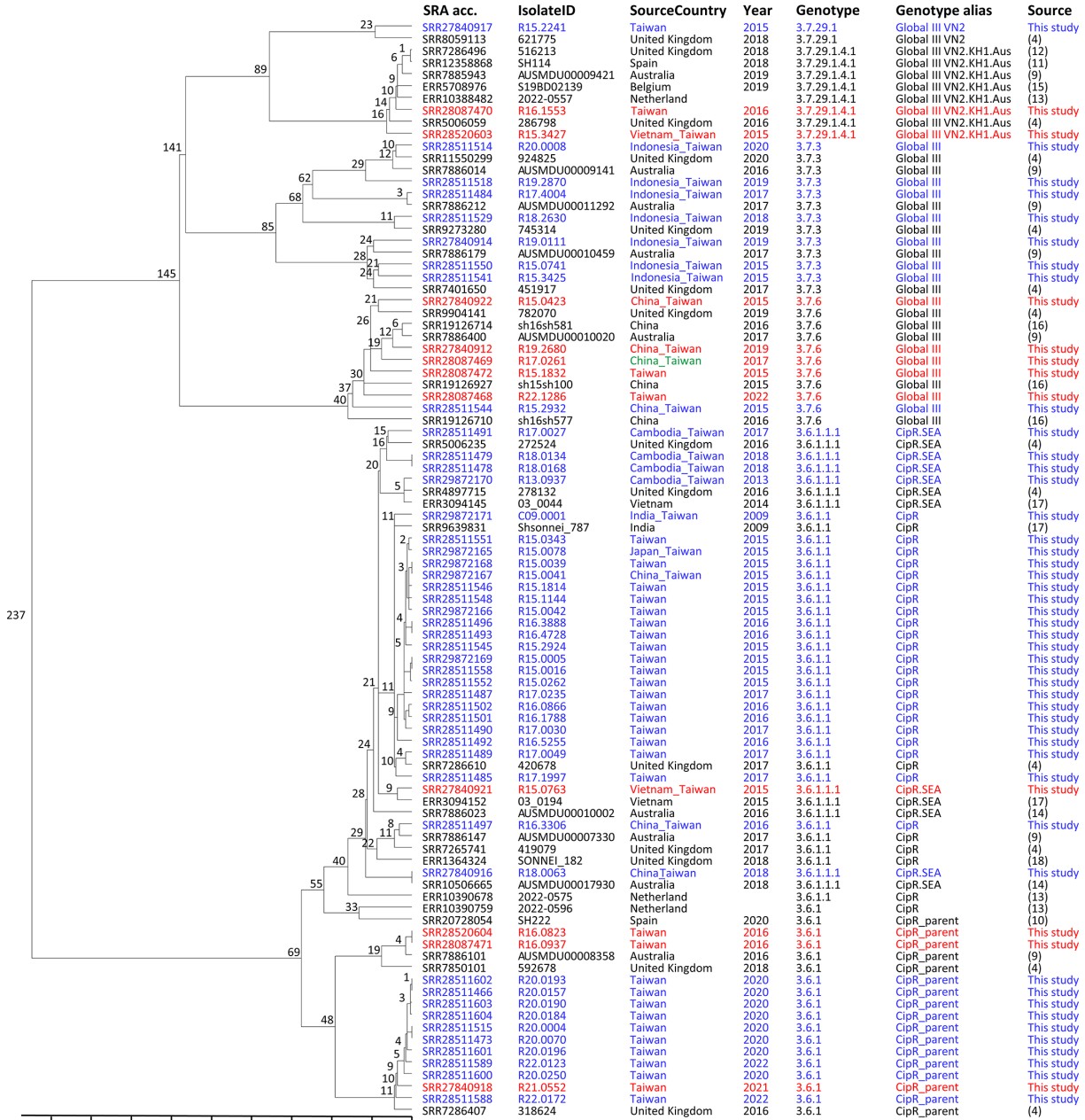

**FIG 3** Phylogenetic tree of *Shigella sonnei* lineage III (SSIII) isolates from Taiwan and other countries. The tree was constructed using cgSNP profiles with the single-linkage algorithm, using *S. sonnei* strain 53G as the reference for SNP calling. Taiwan isolates are highlighted in blue, and XDR isolates from Taiwan are highlighted in red.

caused by non-MSM-associated genogroups (SF2, SSIII, and others) were predominantly imported (82.1%) and mostly occurred in females (68.1%). Of the imported cases, 78.2% (305/390) were from Indonesia, with 80.7% (246/305) of these being female, and none were infected by the three MSM-associated genogroups (Table S6). Most of these Indonesians were female migrant workers employed as caregivers in Taiwan.

Overall, 75.2% of *Shigella* isolates from 2015 to 2022 were MDR (Table 1). AziR_SF3, SF2, and SSIII exhibited high MDR rates, ranging from 85.7% to 100%. MSM-associated genogroups collectively had significantly higher rates of resistance to azithromycin, ESCs, nalidixic acid/ciprofloxacin, and tetracycline compared to non-MSM-associated genogroups (Table 1). Except for nalidixic acid/ciprofloxacin, CipR_SF2 exhibited a higher

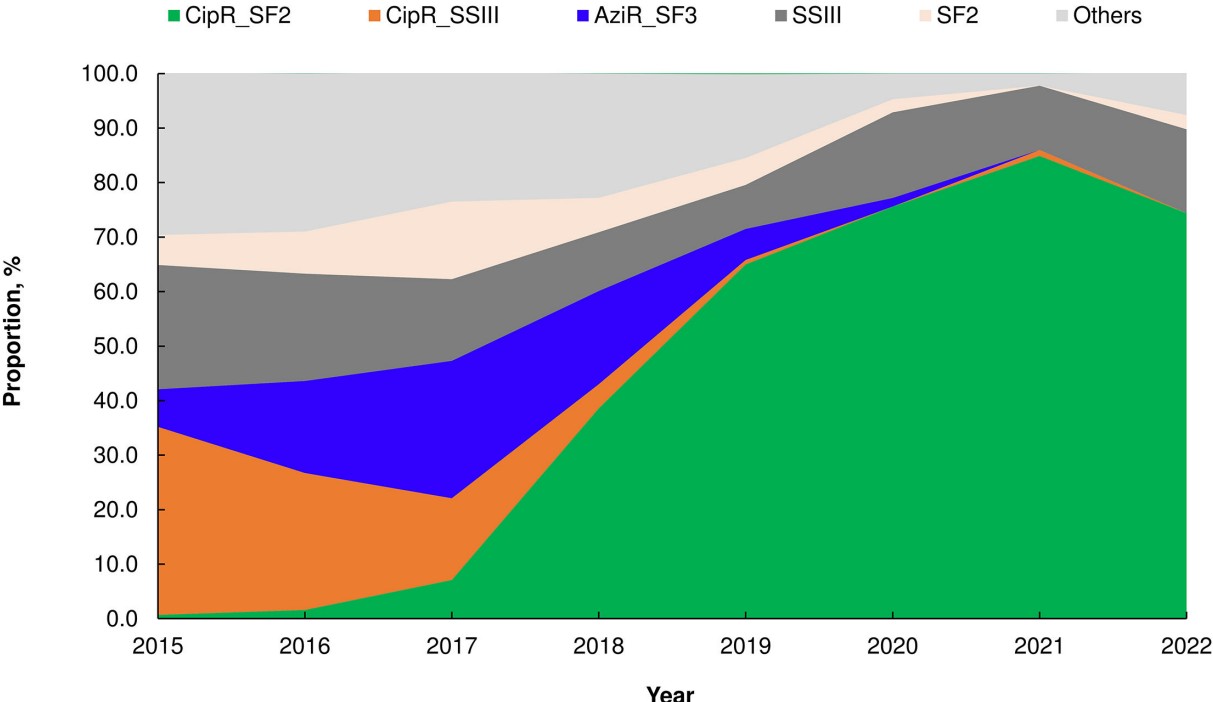

**FIG 4** Distribution of *Shigella* genogroups from 2015 to 2022.

intermediate resistance rate to cefoxitin but lower resistance rates to ampicillin and chloramphenicol, as well as a lower MDR rate compared to SF2. Similarly, CipR_SSIII showed higher rates of resistance to ampicillin, ESCs, and chloramphenicol compared to SSIII. All AziR_SF3 isolates were MDR, with high resistance rates (75.0% to 100%) to ampicillin, chloramphenicol, streptomycin, and tetracycline, but only 40.5% exhibited resistance to azithromycin.

## Genomic characteristics of XDR *Shigella* isolates

Fourteen isolates (11 *S. sonnei* and 3 *S. flexneri*) were designated as XDR, defined by nonsusceptibility to ciprofloxacin and resistance to azithromycin and oxyiminocephalo-sporins (cefotaxime and ceftazidime). Each XDR *S. sonnei* isolate harbored between 6 and 11 plasmids, with resistance genes located within the chromosomes and on two plasmids (Table S7). The resistance genes *blaCTX-M-14*, *blaCTX-M-27*, and *blaDHA-1* for ESCs and *ermB* and *mphA* for azithromycin were found on IncFII or IncB/O/K/Z plasmids (Table S7). The XDR *S. sonnei* isolates belonged to genotypes 3.6.1 (3 isolates), 3.6.1.1 (1 isolate), 3.7.6 (5 isolates), and 3.7.29.1.4.1 (2 isolates). Phylogenetic analysis revealed that these XDR *S. sonnei* isolates were closely related to strains from Taiwan and other countries (Fig. 3). The three XDR *S. flexneri* 2a isolates harbored four or five plasmids and carried *aadA1*, *dfrA1*, *sat2*, and *tet(B)* in their chromosomes and IncB/O/K/Z plasmids, as well as *blaDHA-1*, *dfrA17*, *mph(A)*, *qnrB4*, and *sul1* in IncFII plasmids (Table S7). These XDR isolates were closely related to the other seven CipR_SF2 isolates, differing by no more than 10 SNPs (Fig. 2).

## DISCUSSION

Our study indicates that the increase in the proportion of domestic shigellosis cases since 2015 is associated with infections caused by three MSM-associated *Shigella* gen-ogroups: ciprofloxacin-resistant *S. sonnei* (CipR_SSIII), azithromycin-resistant *S. flexneri* 3a (AziR_SF3), and ciprofloxacin-resistant *S. flexneri* 2a (CipR_SF2) (Fig. 4). Compared to 2003–2014, there was a significant shift in infections from females to males and children

**TABLE 1** Demographic characteristics and antimicrobial resistance in *Shigella* genogroups, 2015–2022

| Characteristics | Total | MSM[a] | Non-MSM[a] | AziR_SF3 | CipR_SF2 | SF2 | CipR_SSIII | SSIII | Others |
|---|---|---|---|---|---|---|---|---|---|
| No. isolates | 1,034 | 620 | 414 | 109 | 387 | 61 | 124 | 158 | 195 |
| Source | | | | | | | | | |
| Domestic, % | 62.3 | 91.9 | 17.9 | 95.4 | 94.6 | 9.8 | 80.6 | 31.6 | 9.2 |
| Imported, % | 37.7 | 8.1 | 82.1 | 4.6 | 5.4 | 90.2 | 19.4 | 68.4 | 90.8 |
| Gender | | | | | | | | | |
| Female, % | 30.6 | 5.5 | 68.1 | 0 | 4.7 | 68.9 | 12.9 | 51.9 | 81.0 |
| Males, % | 69.4 | 94.5 | 31.9 | 100 | 95.3 | 31.1 | 87.1 | 48.1 | 19.0 |
| Phenotypic resistance, % | | | | | | | | | |
| Azithromycin[b] | 6.8 | 8.6[c] | 3.0 | 40.5 | 1.6 | 0 | 7.4 | 5.4 | 1.9 |
| Cefoxitin | 1.9 | 2.0 | 1.7 | 0 | 3.1 | 1.8 | 2.6 | 2.7 | 1.1 |
| Cefoxitin[NS] | 9.5 | 16.4[c] | 2.6 | 1.9 | 40.9[d] | 5.5 | 2.6 | 2.7 | 1.7 |
| Cefotaxime | 6.2 | 7.1 | 4.9 | 0 | 6.3 | 3.3 | 16.2 | 9.7 | 1.5 |
| Ceftazidime | 5.7 | 7.1[c] | 3.7 | 0 | 6.3 | 1.6 | 16.1[e] | 6.5 | 2.1 |
| Ertapenem | 0 | 0 | 0 | 0 | 0 | 0 | 0 | 0 | 0 |
| Nalidixic acid | 58.0 | 82.9[c] | 20.8 | 2.8 | 100[d] | 1.6 | 100[e] | 39.6 | 11.9 |
| Ciprofloxacin | 51.9 | 82.7[c] | 5.9 | 2.8 | 99.7[d] | 1.6 | 100[e] | 3.2 | 9.3 |
| Ciprofloxacin[NS] | 58.7 | 83.1[c] | 22.3 | 3.7 | 100[d] | 6.6 | 100[e] | 39.0 | 14.0 |
| Gentamicin | 1.4 | 0.8 | 2.2 | 0 | 1.0 | 0 | 0.8 | 5.8[e] | 0 |
| Chloramphenicol | 27.6 | 22.5 | 35.2[c] | 100 | 5.5 | 90.2[d] | 7.6[e] | 0.6 | 45.4 |
| Streptomycin | 80.6 | 81.8 | 79.4 | 75.0 | 98.4 | 94.5 | 69.8 | 95.5[e] | 64.8 |
| Sulfamethoxazole | 56.4 | 45.7 | 70.3[c] | 12.5 | 42.5 | 56.3 | 81.3 | 89.1 | 60.6 |
| Trimethoprim-sulfamethoxazole | 56.3 | 47.4 | 68.8[c] | 13.1 | 47.6 | 61.0 | 78.0 | 91.3[e] | 54.5 |
| Tetracycline | 88.5 | 92.9[c] | 81.9 | 99.1 | 98.2 | 95.1 | 70.3 | 90.3[e] | 71.1 |
| Colistin | 1.1 | 1.0 | 1.2 | 1.9 | 0.5 | 0 | 1.7 | 1.3 | 1.5 |
| MDR[f] | 75.2 | 75.0 | 75.6 | 100 | 67.2 | 93.4[d] | 78.0 | 85.7 | 61.9 |

[a]MSM includes AziR_SF3, CipR_SF2, and CipR_SSIII; non-SMS includes SF2, SSIII, and others.
[b]Azithromycin resistance in the AziR_SF3 group was observed in 94.1% of isolates from 2015 to 2016 but declined to 26.9% from 2017 to 2020 due to the loss of erm(B) and mph(A) genes on the pKSR100 plasmid in certain isolates.
[c]MSM vs non-MSM, Chi-square *P*-value < 0.05.
[d]CipR_SF2 vs SF2, Chi-square *P*-value < 0.05.
[e]CipR_SSIII vs SSIII, Chi-square *P*-value < 0.05.
[f]MDR, resistant to ≥3 classes of antimicrobials, including azithromycin, β-lactams (ampicillin, cefoxitin, cefotaxime, or ceftazidime), quinolones (nalidixic acid or ciprofloxacin), gentamicin, chloramphenicol, trimethoprim-sulfamethoxazole, tetracycline, and colistin.

to young adults between 2015 and 2022. This shift was largely attributed to the high proportion of infections caused by the three genogroups, mainly circulating among young adult males.

Shigellosis has become a global concern among MSM populations, characterized by a concerning increase in the transmission of MDR and XDR strains (4, 14, 19–21). In Taiwan, the initial occurrences of ciprofloxacin-resistant *S. sonnei* and azithromycin-non-susceptible *S. flexneri* 3a infections among MSM populations were documented in 2016 (5, 6). In this study, our genomic analysis indicates that the MSM-associated *S. sonnei* (CipR_SSIII) isolates from 2015 belong to Mykrobe genotype 3.6.1.1. Genomic analyses from this study and a previous one revealed that the azithromycin-resistant *S. flexneri* 3a (AziR_SF3) isolates belong to an MSM-associated lineage involved in intercontinental spread (3). Although the ciprofloxacin-resistant *S. flexneri* 2a (CipR_SF2) strains from Taiwan are genetically distant from strains from MSM-associated outbreaks in Australia (9, 14), the Netherlands (13), Spain (10, 11), and the United Kingdom (12) (Fig. 2), they are very likely linked to MSM population in Taiwan, as most cases were domestically acquired (94.6%) and occurred in young adult males (95.3%). CipR_SF2 has predominated in Taiwan since 2018. Noteworthy, closely related CipR_SF2 strains have been found in Australia, the United Kingdom, and France (Fig. 2).

Our genomic analysis reveals that the isolates of *S. sonnei* lineage III from 2015 to 2022 belong to seven Mykrobe genotypes, which have been distributed worldwide (4, 22). The two XDR isolates of genotype 3.7.29.1.4.1 are closely related to strains from

many countries, including Australia, Belgium, the Netherlands, Spain, and the United Kingdom (Fig. 3). Further study indicates that these two isolates are also closely related to the strains from the MSM-associated outbreak that occurred in Montréal, Canada (23), being classified into the SNP cluster PDS000019750.368 by the NCBI Pathogen Detection system. Cluster PDS000019750.368 comprises 963 genomes from at least 10 countries. The five XDR isolates of genotype 3.7.6 are closely related to strains from Australia, China, and the United Kingdom (Fig. 3). The five XDR isolates, including three imported from China, are classified into the SNP cluster PDS000125591.21, comprising 401 genomes, most from China. This XDR clone caused six waterborne shigellosis outbreaks in China from 2015 to 2020 (16). The three XDR isolates of genotype 3.6.1 belong to two subclusters (Fig. 3) and two SNP clusters. These two SNP clusters together comprise 54 genomes from seven countries, including Australia, Belgium, France, Korea, Taiwan, the United Kingdom, and the USA. Except for the three XDR isolates from Taiwan, the remaining 51 are not XDR.

All but one of the CipR_SSIII isolates belonged to genotypes 3.6.1.1 and 3.6.1.1.1. These two genotypes are highly resistant and are widespread across multiple countries and continents. The CipR_SSIII isolates of these two genotypes are part of the SNP cluster PDS000188704.8, which includes 5,572 genomes from 28 countries, with 10.3% of them being XDR.

The first reported outbreak of MSM-associated shigellosis caused by azithromycin-resistant *S. flexneri* serotype 3a occurred in the UK in 2009 (24). Following this, MSM-associated *S. flexneri* 3a (AziR_SF3) strains spread across multiple countries and continents, including Taiwan (3, 5). Genomic analysis reveals that AziR_SF3 isolates typically carry *blaOXA-1*, *catA1*, *aadA1*, and *tet(B)* within the *Shigella*-resistance locus multidrug resistance element, along with *blaTEM-1*, *erm(B)*, and *mph(A)* on the plasmid pKSR100 (3). Consequently, AziR_SF3 isolates are expected to be MDR and to exhibit azithromycin resistance. Our data show that while all AziR_SF3 isolates are MDR, only 40.5% exhibit azithromycin resistance (Table 1). The decline in azithromycin resistance rate can be attributed to the subsequent deletion of *erm(B)* and *mph(A)* from the pKSR100 plasmid following the introduction of the resistant strains. These variant strains subsequently became predominant. Our data indicate that azithromycin resistance was observed in 94.1% of AziR_SF3 isolates from 2015 to 2016, declining to 26.9% from 2017 to 2020 (data not shown). AziR_SF3 isolates belong to SNP cluster PDS000061830.752, which comprises 1,857 genomes from nine countries, including Australia, Belgium, Canada, Ireland, the Netherlands, Spain, Taiwan, the United Kingdom, and the USA. Of these genomes, 26.5% did not carry *erm(B)* and *mph(A)* for azithromycin resistance.

CipR_SF2 is particularly significant as it has been the predominant genogroup responsible for shigellosis in Taiwan since 2018. Genomic analysis reveals that CipR_SF2 isolates are genetically more distant from the strains associated with MSM-associated outbreaks in Australia and European countries (Fig. 2). CipR_SF2 isolates belong to two SNP clusters (PDS000121152.7 and PDS000174037.7), which together comprise only 44 genomes, mostly from Taiwan, with a few from Australia, the United Kingdom, and the USA. All genomes in these two clusters harbor four ARGs (*aadA1*, *dfrA1*, *sat2*, and *tet(B)*), along with S83L and D87N mutations in *gyrA* and S80I in *parC*. However, only three isolates from Taiwan carry resistance determinants to display XDR. As detailed in Table S7, these XDR *S. flexneri* isolates developed resistance to azithromycin and ESCs through the acquisition of an IncFII plasmid carrying five ARGs, including $bla_{DHA-1}$, *dfrA17*, *mph(A)*, *qnrB4*, and *sul1*. Notably, $bla_{DHA-1}$, an AmpC β-lactamase gene, is responsible for resistance to ESCs (e.g., cefotaxime and ceftazidime) and cephamycins (e.g., cefoxitin).

In conclusion, this study highlights a significant epidemiological trend in domestic shigellosis in Taiwan from 2015 to 2022, predominantly affecting the MSM population. The rise in domestic shigellosis is attributed to the emergence and spread of highly resistant clones, particularly CipR_SF2, which has become predominant since 2018. Our findings underscore the urgent need for enhanced surveillance and targeted interven-

tions to control the spread of MDR and XDR *Shigella* strains to mitigate public health impact.

## MATERIALS AND METHODS

### Bacterial isolates and demographic information

Shigellosis is a notifiable disease in Taiwan, requiring hospitals to report cases and submit isolates to the Taiwan Centers for Disease Control (Taiwan CDC). Isolates were confirmed as *Shigella* using the Bruker MALDI Biotyper and slide agglutination with antisera from Denka Seiken Co., Ltd. and Sifin Diagnostics GmbH. Statistical data on reported shigellosis cases from 1996 to 2022 were obtained from the Taiwan National Infectious Disease Statistics System (NIDSS; https://nidss.cdc.gov.tw/en/Home/Index). Demographic information, including sex, age, country of citizenship, country of residence, travel history, and year of onset, was retrieved from the Taiwan National Notifiable Disease Surveillance System with authorization from the Taiwan CDC (IRB 110109).

### Pulsed-field gel electrophoresis

For routine genotyping, isolates were analyzed using the standardized PulseNet PFGE protocol (25). Clustering analysis of PFGE patterns was performed using the tools provided in BioNumerics v6.6 (Applied Maths, Belgium), with parameters set to 1.5% pattern optimization, 0.35% band tolerance, the Dice coefficient, and the UPGMA algorithm.

### Whole-genome sequencing

Isolates selected for WGS included representatives from *Shigella* species (*S. flexneri*, *S. boydii*, and *S. sonnei*), PFGE clusters of *S. flexneri* and *S. sonnei*, 14 XDR isolates from 2015 to 2022, and two ciprofloxacin-resistant *S. sonnei* isolates that emerged in 2009 and 2013. Isolates resistant to ESCs and azithromycin were preferentially selected within each PFGE cluster. WGS was performed using the Illumina MiSeq and Oxford Nanopore Technologies (ONT) Nanopore MinION sequencing platforms. Illumina reads were assembled using SPAdes v3.15.3 (26). The assembled sequences were analyzed to identify ARGs, resistance-relevant mutations, and plasmid incompatibility types, using AMRFinder v3.11.26 and PlasmidFinder v2.1.6. For ONT sequencing, genomic DNA was prepared using the Rapid Barcoding Kit (SQK-RBK114.24 kit) to generate barcoded sequencing libraries, which were processed on MinION R10.4.1 flow cells. POD5 raw signal data were basecalled to FASTQ sequences using Dorado 0.5.0 with the dna_r10.4.1_e8.2_400bps_sup@v4.3.0 model. The FASTQ sequences were assembled with Illumina reads using Flye v2.9.2 and Plassembler v1.6.0 to generate complete genomic sequences. These sequences were subsequently polished using pypolca v0.3.0 and Polypolish v0.5.0.

### Antimicrobial susceptibility testing

Isolates were tested for antimicrobial susceptibility with a custom-made Sensititre MIC panel or the EUVSEC3 Sensititre MIC panel (TREK Diagnostic Systems Ltd., West Essex, England). The MIC breakpoints for *Enterobacterales*, as defined by the Clinical and Laboratory Standards Institute 33rd edition (2023), were used to interpret the antimicrobial susceptibility testing results for ampicillin, cefoxitin, cefotaxime, ceftazidime, chloramphenicol, ciprofloxacin, colistin, trimethoprim-sulfamethoxazole, gentamicin, nalidixic acid, sulfamethoxazole, and tetracycline. For azithromycin, an MIC of ≥32 mg/L was set to be resistant. Streptomycin MICs of ≥64 mg/L, 32 mg/L, and ≤16 mg/L were set to be resistant, intermediate, and susceptible, respectively.

## Hierarchical clustering of cgMLST and Mykrobe genotypes

Hierarchical clustering of cgMLST genotype assignments, derived from hierarchical clustering of cgMLST for *Escherichia* and *Shigella* isolates (27), were obtained by uploading Illumina raw reads to the Enterobase database (https://enterobase.warwick.ac.uk/). Mykrobe genotype assignments for *S. sonnei* isolates, based on single nucleotide variant-based phylogenetic analysis, were generated by processing Illumina raw reads with Mykrobe v0.13.0 (22).

## Phylogenetic analysis

Phylogenetic relationships among *S. sonnei* and *S. flexneri* were constructed using cgSNP profiles. The genomes of *S. flexneri* 2a str. 2457T and *S. sonnei* 53G served as reference sequences for cgSNP calling in *S. flexneri* and *S. sonnei*, respectively. Briefly, cgSNP profiles of the isolates were generated by aligning genomic sequences to the reference genome using ska.rust 0.3.7, followed by the detection and removal of recombinations using Gubbins 3.3.1. A pairwise SNP distance matrix was then calculated using snp-dists 0.8.2 (https://github.com/tseemann/snp-dists), and a phylogenetic tree was constructed using scipy 1.7.3. The initial phylogenetic analysis included *S. sonnei* and *S. flexneri* isolates from MSM-associated shigellosis outbreaks in Australia (9, 14), Belgium (15), the Netherlands (13), Spain (10, 11), and the United Kingdom (12, 19), as well as those from the studies by Hawkey et al. (22), Mason et al. (4), Qiu et al. (16), and The et al. (17). The most closely related strains were then selected to construct phylogenetic trees with Taiwanese isolates. Additionally, information on closely or clonally related isolates from other countries within the same SNP cluster with Taiwanese isolates was retrieved from the NCBI Pathogen Isolates Browser.

## ACKNOWLEDGMENTS

The authors thank all the hospitals that provided Shigella isolates for this study. This research was funded by the Ministry of Health and Welfare, Taiwan, through grants MOHW110-CDC-C-315-113107, MOHW111-CDC-C-315-123108, and MOHW112-CDC-C-315-133116.

## AUTHOR AFFILIATION

[1]Center for Research, Diagnostics and Vaccine Development, Centers for Disease Control, Ministry of Health and Welfare, Taichung, Taiwan

## AUTHOR ORCIDs

Chien-Shun Chiou ⓘ http://orcid.org/0000-0003-3674-1030

## FUNDING

| Funder | Grant(s) | Author(s) |
| --- | --- | --- |
| Ministry of Health and Welfare (MOHW) | MOHW110-CDC-C-315-113107 | Chien-Shun Chiou |
| Ministry of Health and Welfare (MOHW) | MOHW111-CDC-C-315-123108 | Ying-Shu Liao |
| Ministry of Health and Welfare (MOHW) | MOHW112-CDC-C-315-133116 | Ying-Shu Liao |

## AUTHOR CONTRIBUTIONS

Ying-Shu Liao, Conceptualization, Data curation, Funding acquisition, Investigation, Resources, Writing – original draft | Bo-Han Chen, Data curation, Formal analysis, Methodology, Writing – original draft | Yu-Ping Hong, Formal analysis, Methodology,

Writing – original draft | You-Wun Wang, Data curation, Formal analysis, Methodology, Writing – original draft | Ru-Hsiou Teng, Data curation, Investigation, Writing – original draft | Shiu-Yun Liang, Data curation, Investigation, Writing – original draft | Jui-Hsien Chang, Data curation, Formal analysis, Writing – original draft | Chi-Sen Tsao, Data curation, Writing – original draft | Hsiao Lun Wei, Data curation, Investigation, Writing – original draft | Chien-Shun Chiou, Conceptualization, Data curation, Formal analysis, Funding acquisition, Investigation, Methodology, Project administration, Resources, Software, Supervision, Validation, Visualization, Writing – original draft, Writing – review and editing

## ADDITIONAL FILES

The following material is available online.

### Supplemental Material

**Supplemental figures (Spectrum02290-24-S0001.pdf).** Fig. S1 to S5.
**Supplemental tables (Spectrum02290-24-S0002.xlsx).** Tables S1 to S7.

### Open Peer Review

**PEER REVIEW HISTORY (review-history.pdf).** An accounting of the reviewer comments and feedback.

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
