## [Reviewer comments · Microbiology Spectrum]

Microbiology Spectrum

The rise in domestic shigellosis and the genomic characteristics of *Shigella* clones linked to men who have sex with men in Taiwan, 2015–2022

Ying-Shu Liao, Bo-Han Chen, Yu-Ping Hong, You-Wun Wang, Ru-Hsiou Teng, Shiu-Yun Liang, Jui-Hsien Chang, Chi-Sen Tsao, Hsiao-Lun Wei, and Chien-Shun Chiou

Corresponding Author(s): Chien-Shun Chiou, Taiwan Centers for Disease Control

Review Timeline:

Submission Date:	September 10, 2024
Editorial Decision:	December 17, 2024
Revision Received:	January 2, 2025
Editorial Decision:	January 22, 2025
Revision Received:	January 23, 2025
Accepted:	January 23, 2025

Editor: Rafael Vignoli

Reviewer(s): The reviewers have opted to remain anonymous.

Transaction Report:

DOI: <https://doi.org/10.1128/spectrum.02290-24>

Re: Spectrum02290-24 (The rise in domestic shigellosis and the genomic characteristics of Shigella clones linked to men who have sex with men in Taiwan, 2015–2022)

Dear Dr. Chien-Shun Chiou:

Thank you for the privilege of reviewing your work. Your work has been reviewed by experts in the field and although they consider it to be a good work, it needs some minor modifications to be published in Microbiology Spectrum.

Below I send the comments including some minimal details from me:

Lines 135 to 137: Please put the allele corresponding to blaCMY and blaCTX-M and rewrite the sentence since blaCMY or blaDHA-1 are not ESBL but ampC.

Lines 201 to 204: It would be good to homogenize with other publications (such as <https://doi.org/10.1038/s41467-023-37672-w>) the definition of XDR, since trimethoprim-sulfamethoxazole is not a therapeutic recommendation for shigellosis by the WHO or the IDSA for example, it would be better to refer to XDR for resistance to azithromycin, oxyiminocephalosporins and ciprofloxacin. Lines 204-206: Since the plasmids were identified using Plasmid Finder, how were the plasmids listed in Table S7 as ND determined?

Lines 292-294: As worded, the ESC resistance mechanism in these isolates should be mentioned.

In Table 1: Shouldn't the percentage of azithromycin resistance in the AziR SF3 group be 100%?

Below you will find, instructions from the Spectrum editorial office, and the reviewer comments.

Revision Guidelines

Sincerely,
Rafael Vignoli

Reviewer #2 (Comments for the Author):

This is an important paper exploring the long term epidemiological trends in Taiwan for shigellosis. It is particularly important as it comes from a region without much discussion regarding the sexually transmissible form of the disease and there are rare subtypes involved in disease.

The genomic analyses are relatively basic methodologically and without huge amounts of contextual data from other strains, but they are sufficient to support the conclusions presented, so this is not an issue.

My main criticism is that the raw sex/gender data of patients from whom the strains are derived is not clearly shown. So the epidemiological shift to MSM comes across a little as inferred rather than explicit. I do think the shift will have happened, but the link could be strengthened by presenting patient demographic data e.g. in Figure 1 alongside age for the two time periods.

Comments from the Editor:

1. Lines 135 to 137: Please put the allele corresponding to *bla*CMY and *bla*CTX-M and rewrite the sentence since *bla*CMY or *bla*DHA-1 are not ESBL but ampC.

Reply: Thank you for your comment. We have clarified and revised the sentence as follows: "Some CipR_SF2 isolates carried resistance genes associated with AmpC β -lactamases, such as *bla*CMY-2 and *bla*DHA-1, as well as extended-spectrum β -lactamase (ESBL) genes, including *bla*CTX-M-14 and *bla*CTX-M-55." This revision includes the specific alleles of the resistance genes and correctly classifies *bla*CMY and *bla*DHA-1 as AmpC β -lactamases, not ESBLs.

2. Lines 201 to 204: It would be good to homogenize with other publications (such as <https://doi.org/10.1038/s41467-023-37672-w>) the definition of XDR, since trimethoprim-sulfamethoxazole is not a therapeutic recommendation for shigellosis by the WHO or the IDSA for example, it would be better to refer to XDR for resistance to azithromycin, oxyiminocephalosporins and ciprofloxacin.

Reply: Thank you for your suggestion to align our definition of extensively drug-resistant (XDR) *Shigella* with other publications and clinical guidelines. We acknowledge that trimethoprim-sulfamethoxazole is no longer recommended by the World Health Organization (WHO) or the Infectious Diseases Society of America (IDSA) for treating shigellosis.

To ensure consistency with recent literature, including Mason et al. (2023), and to reflect current therapeutic recommendations, we have revised our definition of XDR *Shigella* to denote resistance to azithromycin, oxyiminocephalosporins, and ciprofloxacin.

Additionally, we recognize the Centers for Disease Control and Prevention (CDC) definition of XDR *Shigella* as strains resistant to all commonly recommended empiric and alternative antibiotics, including azithromycin, ciprofloxacin, ceftriaxone, trimethoprim-sulfamethoxazole, and ampicillin (https://www.cdc.gov/han/2023/han00486.html?utm_source=chatgpt.com).

Revised Text for Lines 201–204: "... were designated as XDR, defined by nonsusceptibility to ciprofloxacin and resistance to azithromycin and oxyiminocephalosporins (cefotaxime and ceftazidime)."

3. Lines 204-206: Since the plasmids were identified using Plasmid Finder, how were the plasmids listed in Table S7 as ND determined?

Reply: Thank you for your query regarding the plasmids listed as ND in Table S7. We would like to clarify that the chromosomes and plasmids were assembled, but for some plasmids, the replicon types (incompatibility types) could not be identified using PlasmidFinder, which is why these plasmids are labeled as ND (not determined).

To address this concern, we have revised the footnote of Table S7 S7 for clarity as follows: "^a ND, the replicon types were not determined using PlasmidFinder."

4. Lines 292-294: As worded, the ESC resistance mechanism in these isolates should be mentioned.

Reply: Thank you for highlighting the need to explicitly mention the ESC resistance mechanism. We agree that providing this detail enhances clarity and aligns with best practices in scientific reporting.

Revised Text for Lines 292–294: "As detailed in Table S7, these XDR *S. flexneri* isolates developed resistance to azithromycin and ESCs through the acquisition of an IncFII plasmid carrying five ARGs, including *bla*_{DHA-1}, *dfrA17*, *mph(A)*, *qnrB4*, and *sul1*. Notably, *bla*_{DHA-1}, an AmpC β-lactamase gene, is responsible for resistance to ESCs (e.g., cefotaxime and ceftazidime) and cephalomycins (e.g., cefoxitin)."

5. In Table 1: Shouldn't the percentage of azithromycin resistance in the AziR_SF3 group be 100%?

Reply: Thank you for your observation regarding the percentage of azithromycin resistance in the AziR_SF3 group. We understand the expectation for this group to exhibit 100% resistance to azithromycin due to its designation. However, our data reveal a decline in azithromycin resistance over time.

From our analysis, azithromycin resistance was observed in 94.1% of AziR_SF3 isolates from 2015 to 2016, but this rate decreased to 26.9% from 2017 to 2020, even though these isolates remained genetically closely related. This decline can be attributed to the subsequent deletion of the

erm(B) and *mph(A)* genes from the plasmid *pKSR100*, which are responsible for azithromycin resistance. These variant strains have since become predominant. To address this inconsistency, we propose adding a clarifying footnote to Table 1:

Revised Footnote for Table 1: "***Azithromycin resistance in the AziR_SF3 group was observed in 94.1% of isolates from 2015 to 2016 but declined to 26.9% from 2017 to 2020 due to the loss of *erm(B)* and *mph(A)* genes on the pKSR100 plasmid in certain isolates."

Reviewer #2 (Comments for the Author):

1. This is an important paper exploring the long term epidemiological trends in Taiwan for shigellosis. It is particularly important as it comes from a region without much discussion regarding the sexually transmissible form of the disease and there are rare subtypes involved in disease.

Reply: Thank you for your positive feedback and recognition of the importance of this study.

2. The genomic analyses are relatively basic methodologically and without huge amounts of contextual data from other strains, but they are sufficient to support the conclusions presented, so this is not an issue.

Reply: Thank you for your comment. We are glad the genomic analyses were deemed sufficient to support the conclusions and appreciate your understanding of the study's focus on epidemiology and resistance trends in Taiwan.

3. My main criticism is that the raw sex/gender data of patients from whom the strains are derived is not clearly shown. So the epidemiological shift to MSM comes across a little as inferred rather than explicit. I do think the shift will have happened, but the link could be strengthened by presenting patient demographic data e.g. in Figure 1 alongside age for the two time periods.

Reply: Thank you for your valuable feedback. We have addressed your concern by incorporating sex data into Figure 1 to provide a clearer representation of the epidemiological shift alongside age data for the two time periods. Additionally, we have revised the description in the manuscript to read: "*The age and sex distribution of domestic shigellosis cases changed significantly between the two periods. From 2003 to 2014,*

the majority of domestic cases occurred in children (ages 0–9), with males accounting for a smaller proportion of cases across all age groups (Figure 1A). In contrast, from 2015 to 2022, domestic cases were predominantly observed in young adults (ages 20–39), with males comprising a substantially higher proportion of cases across all age groups, particularly among young adults (Figure 1B)."

We hope these updates strengthen the link to the MSM population as suggested.

Re: Spectrum02290-24R1 (The rise in domestic shigellosis and the genomic characteristics of Shigella clones linked to men who have sex with men in Taiwan, 2015–2022)

Dear Dr. Chien-Shun Chiou:

Thank you for the privilege of reviewing your work. Congratulations, all suggestions have been incorporated and the paper will be approved after minor correction.

Please change cephalomycins to cephamycins on line 315.

Revision Guidelines

Sincerely,
Rafael Vignoli
Editor
Microbiology Spectrum

Comment from the Editor:

1. Please change cephalomycins to cephamycins on line 315.

Response: We have corrected "cephalomycins" to "cephamycins" on line 315 as requested.

Re: Spectrum02290-24R2 (The rise in domestic shigellosis and the genomic characteristics of Shigella clones linked to men who have sex with men in Taiwan, 2015–2022)

Dear Dr. Chien-Shun Chiou:

Congratulations! Your manuscript has been accepted, and I am forwarding it to the ASM production staff for publication. Your paper will first be checked to make sure all elements meet the technical requirements. ASM staff will contact you if anything needs to be revised before copyediting and production can begin. Otherwise, you will be notified when your proofs are ready to be viewed.

Sincerely,
Rafael Vignoli
Editor
Microbiology Spectrum